# A Novel Drug Delivery System: Hyodeoxycholic Acid-Modified Metformin Liposomes for Type 2 Diabetes Treatment

**DOI:** 10.3390/molecules28062471

**Published:** 2023-03-08

**Authors:** Minghao Hu, Tingting Gou, Yuchen Chen, Min Xu, Rong Chen, Tao Zhou, Junjing Liu, Cheng Peng, Qiang Ye

**Affiliations:** 1State Key Laboratory of Southwestern Chinese Medicine Resources, Chengdu University of Traditional Chinese Medicine, Chengdu 611137, China; 2College of Pharmacy, School of Modern Chinese Medicine Industry, Chengdu University of Traditional Chinese Medicine, Chengdu 611137, China

**Keywords:** type 2 diabetes, deoxycholic acid, metformin, liposomes

## Abstract

Metformin is a first-line drug for the clinical treatment of type 2 diabetes; however, it always leads to gastrointestinal tolerance, low bioavailability, short half-life, etc. Liposome acts as an excellent delivery system that could reduce drug side effects and promote bioavailability. Hyodeoxycholic acid, a cholesterol-like structure, can regulate glucose homeostasis and reduce the blood glucose levels. As an anti-diabetic active ingredient, hyodeoxycholic acid modifies liposomes to make it overcome the disadvantages of metformin as well as enhance the hypoglycemic effect. By adapting the thin-film dispersion method, three types of liposomes with different proportions of hyodeoxycholic acid and metformin were prepared (HDCA:ME-(0.5:1)-Lips, HDCA:ME-(1:1)-Lips, and HDCA:ME-(2:1)-Lips). Further, the liposomes were characterized, and the anti-type 2 diabetes activity of liposomes was evaluated. The results from this study indicated that three types of liposomes exhibited different characteristics—Excessive hyodeoxycholic acid decreased encapsulation efficiency and drug loading. In the in vivo experiments, liposomes could reduce the fasting blood glucose levels, improve glucose tolerance, regulate oxidative stress markers and protect liver tissue in type 2 diabetic mice. These results indicated that HDCA:ME-(1:1)-Lips was the most effective among the three types of liposomes prepared and showed better effects than metformin. Hyodeoxycholic acid can enhance the hypoglycemic effect of metformin and play a suitable role as an excipient in the liposome.

## 1. Introduction

Diabetes is a common chronic disease. According to the International Diabetes Federation, the number of people with diabetes has reached 463 million globally in 2019. It is estimated that the number of patients with diabetes will increase to 578 million by 2030 [1]. Diabetes is a kind of disease that occurs when blood glucose levels are high and is caused by insufficient insulin secretion or impaired insulin action. Diabetes is classified as type 1 diabetes and type 2 diabetes [2,3]. Type 2 diabetes accounts for more than 90% of diabetic cases [4]. Sustained high blood glucose levels can damage the eye, kidney, cardiovascular and nervous systems, etc. Diabetes usually results in many complications, such as diabetic nephropathy, diabetic neuropathy, and diabetic foot [5]. These complications often lead to blindness and disability, even death, thereby seriously endangering human health.

Metformin has remarkable hypoglycemic effects and is used as a first-line drug for the treatment of type 2 diabetes. Although metformin could lead to side effects in the gastrointestinal system (nausea, diarrhea, etc.), clinical trials have proved few safety concerns for metformin. Moreover, long-term oral administration increases the risk of lactic acidosis [6]. Due to the transmembrane difficulty, low bioavailability and short half-life in vivo, metformin is difficult to be fully utilized clinically [7,8]. Developing sustained-release preparations is an effective way for improving bioavailability and reducing side effects.

Liposome is composed of a lipid bilayer and can encapsulate hydrophilic and lipophilic drugs. Liposomes are similar in nature and function to the biological membrane and can deliver drugs to the cell interior by fusing with cell membranes [9]. Compared with traditional drug delivery systems, liposomes exhibit outstanding characteristics of high drug loading, sustained-release, low toxicity, biocompatibility and biodegradability [10]. Appropriate active ingredients added to liposomes not only play the role of excipients but also enhance the efficacy of the main drug [11]. In Peony and Licorice stomach floating tablets, chitosan as excipients could assist the formation of drug tablets and promote the healing of gastric ulcer wounds [12,13]. Thus, it is feasible to design a new drug delivery system based on the above strategy [14]. A previous study indicated that the nanoparticles modified by ginsenoside could enhance the anti-tumor effect of drugs, reduce the drug side effects and improve bioavailability and targeting [15]. Moreover, a nanostructured lipid carrier with naringin-containing coix seed oil demonstrated synergistic anti-tumor activities against hepatocellular carcinoma [16]. In several new drug delivery systems, liposomes have received special attention.

In ancient China, pig bile was used for diabetes treatment [17]. A modern study also suggested that hyocholic acid and its derivatives were closely related to diabetes and could be considered new biomarkers for diabetes [18]. Further studies had found that hyodeoxycholic acid (HDCA) up-regulated the production and secretion of glucagon-like peptide-1 (GLP-1) in intestinal endocrine cells by simultaneously activating the G-protein-coupled BA receptor TGR5 and inhibiting the Fani-like X receptor (FXR), thereby inducing the secretion of insulin and reducing blood glucose levels [19]. Thus, it was revealed that HDCA could regulate glucose homeostasis and reduce blood glucose levels. To achieve excellent hypoglycemic effects and cholesterol-like structure, cholic acid and its derivatives can also be used as excipients for liposome modification, which would enhance the efficacy, bioavailability, and targeted delivery of the main drug. Chen et al. prepared cholic acid-modified biphenyl diester liposomes and observed that liposomes modified by cholic acid could enhance drug accumulation in the liver parenchymal cells [20]. Tan demonstrated that the anti-cerebral ischemia activity of baicalin liposomes modified with cholic acid was significantly higher than that of ordinary baicalin liposomes, indicating the enhancement of the therapeutic of baicalin by cholic acid [21]. Therefore, based on the findings from the above studies, HDCA-modified liposomes containing metformin are used to treat type 2 diabetes and are found to be better than using the metformin alone.

In this study, HDCA was used as an excipient for the modification of liposomes. Three different liposomes (HDCA:ME-(0.5:1)-Lips, HDCA:ME-(1:1)-Lips, and HDCA:ME-(2:1)-Lips) were prepared by the thin-film dispersion method. Further, the particle size and polydispersion index (PDI), encapsulation rate (EE), and drug loading (DL) of liposomes were determined. The activities of the three different proportions of liposomes against type 2 diabetes in vivo were sequentially evaluated by measuring fasting blood glucose, glucose tolerance, biochemical markers, and pathological tissue.

## 2. Results

### 2.1. Preparation and Characterization of Liposomes

Three types of HDCA-modified metformin liposomes (HDCA:ME-(0.5:1)-Lips, HDCA:ME-(1:1)-Lips, and HDCA:ME-(2:1)-Lips) were prepared by the thin-film dispersion method [22]. To evaluate the stability of liposomes, the particle size and PDI were determined. The particle sizes of the three liposomes were about 196.36 nm, 232.30 nm, and 332.40 nm, respectively, and the PDI of all three liposomes was smaller than 0.30 (Table 1). The EE of the liposomes was 67.15%, 61.28% and 59.53%, and the DL was 5.52%, 5.35%, and 5.16%, respectively (Table 1). The particle size of HDCA:ME-(0.5:1)-Lips was less than 200 nm, while those of HDCA:ME-(1:1)-Lips and HDCA:ME-(2:1)-Lips were greater than 200 nm, indicating that HDCA:ME-(0.5:1)-Lips could easily pass through the membrane and escape physical removal [23]. Moreover, EE and DL gradually decreased with the increase in HDCA dosage, indicating the importance of HDCA dosage for EE and DL of liposomes.

### 2.2. Assay for Anti-Type 2 Diabetes Activity

#### 2.2.1. Reduction in the Fasting Blood Glucose Level

The effects of different liposomes on the fasting blood glucose levels in 12 h fasting mice are shown in Table 2.

As shown in Table 2, the fasting blood glucose levels of the model group mice ranged from 18.26 to 20.48 mmol/L. The blood glucose levels of the model group were significantly higher (*p* < 0.05) than those of the normal group, indicating that the models of type 2 diabetic mice were successfully established. The fasting blood glucose levels of the liposome groups and the metformin group significantly decreased from day 7 to day 21 (*p* < 0.05, *p* < 0.01). Considering oral administration of metformin for 21 days, the blood glucose levels of the HDCA:ME-(0.5:1)-Lips group were close to that of the metformin group; the levels of the HDCA:ME-(1:1)-Lips group were slightly lower than those of the metformin group; and the hypoglycemic effect of HDCA:ME-(2:1)-Lips group was higher than that of the metformin group. Thus, the hypoglycemic effect of the HDCA:ME-(1:1)-Lips group was slightly better than the metformin group.

The body weight of mice from six groups showed no significant difference after 21 days of administration (Figure 1a). The metformin group significantly reduced the food intake of type 2 diabetic mice after 2 weeks of treatment (*p* < 0.01) (Figure 1b). After 3 weeks of administration, the food intake of the three liposome groups decreased significantly when compared with the model group. It was observed that a slight decrease in food intake in the HDCA:ME-(0.5:1)-Lips group after the treatment of 3 weeks. However, there was no significant difference between the HDCA:ME-(1:1)-Lips and HDCA:ME-(2:1)-Lips groups.

#### 2.2.2. Improvement in the Oral Glucose Tolerance

In this study, an oral glucose tolerance test was performed. After the administration of glucose, the blood glucose levels of the mice were measured at different time intervals (Figure 2).

The results demonstrated that the blood glucose levels of all the mice increased rapidly within 30 min, and then gradually decreased from 30 min to 120 min. The blood glucose levels of the model group were significantly higher than those of normal group (*p* < 0.05). The blood glucose levels of the liposome groups and the metformin group significantly decreased (*p* < 0.05) when compared with the model group. Between 30 and 120 min, the blood glucose levels of the HDCA:ME-(0.5:1)-Lips and HDCA:ME-(2:1)-Lips group were higher than those of the metformin group, while the blood glucose levels of the HDCA:ME-(1:1)-Lips group were lower than those of the metformin group. The efficacy of the HDCA:ME-(1:1)-Lips group was slightly better than the metformin group. In three liposome groups, the time of effect in the HDCA:ME-(1:1)-Lips group was significantly faster than those in HDCA:ME-(0.5:1)-Lips and HDCA:ME-(2:1)-Lips group. The analysis of blood glucose at different time intervals, indicated that the improvement effect of HDCA:ME-(1:1)-Lips on glucose tolerance was significantly higher than that of metformin in type 2 diabetic mice. Furthermore, a flattening hypoglycemic trend of the HDCA:ME-(1:1)-Lips group was observed between 90 and 120 min, revealing that HDCA:ME-(1:1)-Lips group demonstrated a certain sustained-release effect.

#### 2.2.3. Regulation of the Biochemical Indices

Total cholesterol (TC) and triglyceride (TG) are important clinical indicators of blood lipid [24]. The pathological elevation of TC increases the risk of hyperlipoproteinemia, atherosclerosis, and diabetes [25]. High levels of TG can lead to atherosclerosis, diabetes, and pancreatitis [26]. As shown in Figure 3a,b, the TC and TG of the model group were higher than those of the normal group (*p* < 0.05). The high levels of TC and TG were reduced in the metformin group and the liposome groups (*p* < 0.05). The TC levels in the HDCA:ME-(1:1)-Lips group were higher than those in the HDCA:ME-(0.5:1)-Lips and HDCA:ME-(2:1)-Lips groups. The TG levels in the three HDCA-ME-Lips groups were higher than those in the metformin group. Thus, HDCA:ME-(1:1)-Lips group performed the best.

As type 2 diabetes is characterized by insulin resistance, insulin (INS) levels can reflect the effectiveness of drugs [27]. As shown in Figure 3c, there were a significant increase in the serum INS levels of the model group (*p* < 0. 01), indicating that type 2 diabetic mice were successfully modeled. There was a reduction in the INS levels in the metformin group and the three liposome groups reduced the levels of INS (*p* < 0.05) when compared with the model group. The INS levels in the HDCA:ME-(1:1)-Lips group were significantly higher than those in the metformin group. However, the INS levels in the HDCA:ME-(0.5:1)-Lips and HDCA:ME-(2:1)-Lips groups were lower than those in the metformin group. These results demonstrated that the HDCA:ME-(1:1)-Lips group could better increase INS sensitivity and improve the symptoms of INS resistance in type 2 diabetic mice. Glucagon-like peptide 1 (GLP-1) is considered to be a key factor in postprandial glucose homeostasis regulation [28]. It was reported that metformin and bile acid can significantly increase GLP-1 secretion [29]. As shown in Figure 3d, significant increases in the GLP-1 levels in the metformin group and the three liposome groups were observed. The GLP-1 levels in three liposome groups were obviously higher than those in the metformin group.

Superoxide dismutase (SOD), catalase (CAT), and malonaldehyde (MDA) are important indicators of oxidative stress and play a significant role in glucose metabolism [30,31,32,33]. The activity of SOD and CAT was significantly reduced in the model group when compared with that in the normal group (*p* < 0.01) (Figure 3d–f). When diabetic mice were treated with metformin and three types of liposomes, the activity of SOD and CAT increased significantly (*p* < 0.05). When compared with the metformin group, the activity of SOD and CAT was enhanced in the HDCA:ME-(1:1)-Lips group. However, the activity of SOD and CAT in the HDCA:ME-(0.5:1)-Lips and HDCA:ME-(2:1)-Lips groups were lower than that in the metformin group. The MDA content of the model group was much higher than that of the normal group (*p* < 0.01). The MDA content of the metformin group and the liposome groups were effectively reduced when compared with that of the model group. Moreover, the reduction in MDA content of the HDCA:ME-(1:1)-Lips group was significantly better than that of the metformin group. These results indicated that HDCA-ME-Lips may relieve oxidative stress and abnormal glucose metabolism caused by oxidative stress, thereby regulating the oxidative decomposition of glucose to decrease blood glucose.

The effects of three types of liposomes on different biochemical indices were compared in Table 3. The TC and TG levels of the HDCA:ME-(1:1)-Lips group were lower than those of the HDCA:ME-(0.5:1)-Lips and HDCA:ME-(2:1)-Lips groups. The down-regulation in INS levels and up-regulation in GLP-1 levels of the HDCA:ME-(1:1)-Lips group was significantly better than those of the HDCA:ME-(0.5:1)-Lips and HDCA:ME-(2:1)-Lips groups. Furthermore, we also observed that the SOD, CAT, and MDA levels in the three liposome groups there existed a significant difference. The activity of SOD and CAT in the HDCA:ME-(1:1)-Lips group was higher than that in the HDCA:ME-(0.5:1)-Lips and HDCA:ME-(2:1)-Lips groups. However, the MDA content of the HDCA:ME-(1:1)-Lips group was lower than that of the HDCA:ME-(0.5:1)-Lips and HDCA:ME-(2:1)-Lips groups. HDCA:ME-(1:1)-Lips showed stronger than HDCA:ME-(0.5:1)-Lips and HDCA:ME-(2:1)-Lips.

#### 2.2.4. Protection of the Liver Tissue

The results of the histopathological examination revealed significant differences in the morphology and the number of mice hepatocytes in the different experimental groups (Figure 4). The hepatocytes of the normal group had clear boundaries and round nuclei, which were surrounded by rich cytoplasm (Figure 4a). The model group exhibited infiltration of inflammatory cells, cell swelling, focal necrosis, and cytoplasm osteoporosis. The cells appear translucent and polygonal-shaped (Yellow arrow) (Figure 4b). The local necrosis and lymphocyte infiltration in type 2 diabetic mice hepatocytes were significantly relieved in the metformin group and the liposomes groups (Figure 4c–f). In HDCA:ME-(0.5:1)-Lips and HDCA:ME-(2:1)-Lips groups, the intercellular lacuna was significantly enlarged, the cytoplasm was clear, and the nucleus was atrophic. In the HDCA:ME-(1:1)-Lips group, there were fewer swollen and necrotic cells and stickier cytoplasm. Moreover, the cell shape is more regular, and the nuclei are arranged more neatly. There were fewer symptoms of lymphocyte infiltration and focal necrosis in the HDCA:ME-(1:1)-Lips group than those in the metformin group, indicating that HDCA:ME-(1:1)-Lips effectively alleviated inflammation and protected the tissue structure of the liver in type 2 diabetic mice.

## 3. Discussion

In this study, three different proportions HDCA-modified ME liposomes were prepared. The study evaluating the effects of liposomes on mice with type 2 diabetes came up with the following conclusions.

### 3.1. Effect of HDCA Dosage on Liposomes

As a hydrophilic small molecular drug, metformin can be encapsulated by liposomes [34,35]. Three types of HDCA-modified metformin liposomes were prepared by the thin-film dispersion method, which was simple and reproducible. The characterization of the three types of liposomes indicated that the dosage of HDCA is an important factor governing the morphology of liposomes. The particle size of a liposome determines its ability to penetrate biofilms. The smaller particle sizes of the liposomes indicate stronger penetration ability, which is favorable for the drugs to reach target organs or target cells [36]. Moreover, the effect of HDCA dosage on EE and DL should not be ignored. As shown in Table 1, EE and DL of the liposomes gradually decreased with the increase in the HDCA dosage. It is worth noting that the more HDCA dosage in the preparation of liposomes indicated thicker film formation, which makes the preparation of liposomes more difficult. These results indicated that excessive excipients would reduce EE and DL, which in turn, would affect the efficacy of the liposomes to a certain extent. Therefore, choosing appropriate excipients and drugs is the key to the successful preparation of liposomes.

### 3.2. Anti-Type 2 Diabetes Activity of Liposomes

Further, the effect of HDCA-modified metformin liposomes on type 2 diabetes mice was investigated. Liposomes significantly decreased the fasting blood glucose levels in type 2 diabetic mice (*p* < 0.05) and improved oral glucose tolerance (*p* < 0.05). TC and TG of the diabetic patients were maintained at high levels, which may lead to cardiovascular complications [37,38]. Liposomes significantly reduced TC and TG levels in type 2 diabetic mice (*p* < 0.05). INS resistance is a characteristic of type 2 diabetes [39]. The INS levels were significantly decreased by the liposomes (*p* < 0.05), indicating that liposomes could better increase insulin sensitivity and improve the symptoms of INS resistance in type 2 diabetic mice. The liposomes also can increase the secretion of GLP-1, which is a crucial factor for postprandial glucose homeostasis. SOD, CAT, and MDA are oxidative stress indicators, which can reflect glucose metabolism in several ways [40,41]. Liposomes significantly increased SOD and CAT activity and decreased MDA content in the liver, demonstrating that liposomes could regulate glycolipid metabolism and reduce blood sugar by relieving oxidative stress to a certain extent. The histopathological examination demonstrated that lymphocyte infiltration and local necrosis were reduced by liposomes, indicating an effective reduction in inflammation and protection of liver tissue structure in type 2 diabetic mice. This helped control the progression of diabetes and related complications.

Among three types of liposomes, HDCA:ME-(1:1)-Lips significantly reduced the fasting blood glucose levels, improved glucose tolerance, and regulated biochemical indices (TC, TG, INS, GLP-1, SOD, CAT, and MDA) in type 2 diabetic mice. The anti-diabetic effect of HDCA:ME-(1:1)-Lips was also significantly better than that of metformin, HDCA:ME-(1:1)-Lips and HDCA:ME-(1:1)-Lips. In addition, HDCA:ME-(1:1)-Lips was found to protect liver tissue better than metformin (Figure 4f). Therefore, we speculated that HDCA achieved synergistic hypoglycemic effect in the liposomes.

### 3.3. Potential of HDCA as an Excipient for Liposomes

HDCA, a steroid derivative, is similar in structure to cholesterol [42]. HDCA as an excipient can replace cholesterol for the stabilization of the phospholipid bilayer. The biparental nature of HDCA makes it possible to modify liposomes [43]. It is noteworthy that HDCA could regulate glucose homeostasis and reduce blood glucose, which was considered a potential natural ingredient in the treatment of type 2 diabetes [44]. In this study, as an excipient for liposome modification, HDCA has strong hypoglycemic activity and can be used for the treatment of type 2 diabetic mice synergistically with metformin. On the other hand, HDCA can reduce the destruction of metformin liposomes in the gastrointestinal tract and improve the stability of liposomes in vivo. In addition, HDCA-modified metformin liposomes may reduce the side effects of metformin as well as improve its bioavailability and half-life. However, this still needs to be verified in future experiments.

In conclusion, HDCA in liposomes plays a very crucial role as excipients and auxiliary anti-type 2 diabetes ingredients and has a significant potential for the treatment of type 2 diabetes. This strategy of liposomes modified with natural anti-diabetic active ingredients is expected to be applied to other type 2 diabetes drugs for improving their hypoglycemic efficacy and reducing their side effects.

## 4. Materials and Methods

### 4.1. Materials

Metformin hydrochloride and hyodeoxycholic acid were purchased from Shanghai Yi En Chemical Technology Co., Ltd. (Shanghai, China); Lecithin was bought from Chengdu Kelon Chemical Co., Ltd. (Chengdu, China). The assay kits for total cholesterol (TC), total triglycerides (TG), superoxide dismutase (SOD), catalase (CAT), and malonaldehyde (MDA) were purchased from Nanjing Jiancheng Bioengineering Institute (Nanjing, China). The insulin (INS) and Glucagon-like peptide 1 (GLP-1) assay kit was purchased from Wuhan ABclonal Biotechnology Co., Ltd. (Wuhan, China). Acetonitrile was HPLC grade, and all other reagents were analytical grade. Purified deionized water was used throughout the experiment.

### 4.2. Preparation of Liposomes

Liposomes were prepared by the thin-film dispersion method. Certain proportions of lecithin and HDCA were precisely weighed in a pear-shaped bottle and dissolved in 10 mL of trichloromethane-methanol (3:1) solution using ultrasonication. Subsequently, the mixed solution was placed on a rotary evaporator (RE-52AA, ShangHai Yarong Biochemistry Instrument Factory, Shanghai, China), and the organic solvent was removed under a vacuum at 40 °C. When a uniform film was formed on the bottle, a 10 mL PBS solution of metformin (10 mg/mL) was added. Then the film was hydrated at 40 °C for 1 h. The liposome suspension was ultrasonicated for 10 min for uniform dispersion. Then, the suspension was filtered through 0.45 and 0.22 μm microporous membranes and stored at 4 °C. 100 mg of metformin was added to 50 mg, 100 mg, and 200 mg HDCA to obtain three different proportions of liposomes, namely, HDCA:ME-(0.5:1)-Lips, HDCA:ME-(1:1)-Lips, and HDCA:ME-(2:1)-Lips.

### 4.3. Characterization of Liposomes

#### 4.3.1. Particle Size and Polydispersity Index (PDI)

The particle sizes and polydispersity index (PDI) of liposomes were determined using a nanoparticle size analyzer (Litesizer 500, Anton Paar, Graz, Austria). The liposomes were diluted 20-fold with purified water before measurement, and each measurement was repeated three times.

#### 4.3.2. Encapsulation Efficiency (EE) and Drug Loading (DL)

One mL of liposome was centrifuged for 30 min at 12,000 r/min. The supernatant was collected and was placed in a 10 mL bottle. The anhydrous ethanol was used for constant volume and the solution was mixed by vortex for 5 min. The solution was further diluted with anhydrous ethanol to obtain the desired concentration. The absorbance of metformin was determined by the ultraviolet/visible spectrophotometer (UV-2550, Shimadzu, Kyoto, Japan) at 232 nm. The concentration of free metformin (C_free_) was calculated using the following standard curve: y = 0.0758x + 0.1413 (R^2^ = 0.9998; *n* = 6). The standard curve was constructed using a series of standard metformin solutions with concentrations ranging from 4.65 µg/mL to 74.4 µg/mL. The total metformin concentration (C_total_) was determined by adding another 1 mL liposome in a 10 mL volumetric bottle. Further, 4 mL of DMSO was added to the bottle and the volume was made up by anhydrous ethanol. The solution was then mixed by vortex for 5 min. Then solution was further diluted with anhydrous ethanol to obtain the desired concentration. Equations (1) and (2) were used to calculate the EE and DL of the liposomes, respectively.
(1)$$\;\mathrm{EE}\%=\frac{\;\mathrm{Ctotal}-\mathrm{Cfree}}{\;\mathrm{Ctotal}}\times 100\%$$
(2)$$\mathrm{DL}\%=\frac{\mathrm{encapsulated}\;\mathrm{drug}\;\mathrm{content}}{\mathrm{weight}\;\mathrm{of}\;\mathrm{carrier}}\times 100\%$$

### 4.4. Animal Experiments

KM mice (18–22 g, male) were purchased from Spearfish (Beijing, China) Biotechnology Co. All mice were acclimatized to the new environment for one week, during which they were kept in light for 12 h and dark for 12 h per day, with appropriate diet control and free access to water. Mice were prohibited from ingesting food for 12 h before the experiment, but drinking water was not restricted. All the animal experimental protocols were approved by the Animal Research Ethics Committee of Chengdu University of Traditional Chinese Medicine (NO. 2022-55).

#### 4.4.1. Establishment of the Type 2 Diabetic Mice Model

One hundred mice were reared for 7 days before the experiment. The blood glucose levels of mice were measured with a glucose meter (yuyue glucometer, Shenzhen, China) before preparation for modeling. No abnormalities in mice were observed. Mice were randomly divided into the model group and the normal group, then were given a normal diet and distilled water. There were ninety mice in the model group and ten mice in the normal group. Mice of the model group were fasted for 12 h and injected intraperitoneally with 0.4% streptozotocin (STZ) solution (0.2 mL per mouse with a corresponding dose of 50 mg/kg) for 5 days. On the fifth day after injection, food and water were normally provided. The fasting blood glucose levels were weekly measured. Since this method requires stable blood glucose levels for 1–2 weeks, fasting blood glucose levels were measured during the second week. The mice having blood glucose levels between 11.1 and 20 mmol/L were selected as the successful type 2 diabetes mice.

#### 4.4.2. Grouping and Administration

According to the fasting blood glucose levels, thirty STZ-treated type 2 diabetes mice were randomly divided into five groups with 6 mice in each group. Another six untreated mice were randomly selected as the normal group. Each group was administered the following compounds orally for 21 days: (1) Normal group mice were administered 0.9%NaCl; (2) Model group mice were administered 0.9%NaCl; (3) Metformin group mice were administered metformin hydrochloride at a dose of 100 mg/kg/d; (4) HDCA:ME-(0.5:1)-Lips group mice were administered HDCA:ME-(0.5:1)-Lips at a dose of 100 mg/kg/d; (5) HDCA:ME-(1:1)-Lips group mice were administered HDCA:ME-(1:1)-Lips at a dose of 100 mg/kg/d; and (6) HDCA:ME-(2:1)-Lips group mice were administered HDCA:ME-(2:1)-Lips at a dose of 100 mg/kg/d.

#### 4.4.3. Measurement of Fasting Blood Glucose Levels

Before measurement, the mice were prohibited from ingesting food for 12 h. The blood glucose levels of caudal vein blood were measured by a glucose meter at 0, 7, 14, and 21 days after administration.

#### 4.4.4. Oral Glucose Tolerance Test

After mice were administered with the aforementioned compounds for 21 days, the oral glucose tolerance test was performed. Mice were not allowed to ingest food for 12 h and then orally administered 2.0 g/kg glucose. The blood glucose levels were measured at 0, 30, 60, and 120 min after the glucose administration.

#### 4.4.5. Assessment of Biochemistry Indices

Serum was collected by centrifugation for 10 min at 3000 r/min using a refrigerated centrifuge at 4 °C. Biochemical indices including TG, TC, INS and GLP-1 were measured using the assay kit following the manufacturer’s instructions. According to the ratio of liver tissue weight (g): normal saline (mL) = 1:9, homogenization of liver tissues was performed. The supernatants were collected by centrifugation for 10 min at 8000 r/min at 4 °C. The hepatic SOD, CAT, and MDA were determined normatively using assay kits.

#### 4.4.6. Histopathological Examination

Liver tissues were embedded in paraffin and the paraffin sections were sliced and stained with hematoxylin–eosin (HE). The tissues were visualized by a CX41 microscope (Olympus, Tokyo, Japan) equipped with MDX4 (Lilai, Chengdu, China) digital camera system under 400× magnification.

### 4.5. Statistical Analysis

All the experiments were repeated at least 3 times. All the experimental data were presented as mean ± standard deviation (SD). One-way analysis of variance was performed on the data using SPSS 26.0 software (IBM, Chicago, IL, USA). *p* < 0.05 was considered statistically significant.

## 5. Conclusions

In this study, three HDCA-modified metformin liposomes (HDCA:ME-(0.5:1)-Lips, HDCA:ME-(1:1)-Lips, and HDCA:ME-(2:1)-Lips) were prepared by the thin-film dispersion method based on the different proportions of HDCA and metformin. The three types of liposomes exhibited different characteristics (particle size, EE, and DL), which was related to the dosage of HDCA. When the dosage of HDCA increased, the particle size, EE and DL of liposomes also decreased obviously. In addition, liposomes showed significant anti-diabetic effects in type 2 diabetic mice. Liposomes could reduce the fasting blood glucose levels, improve glucose tolerance and regulate biochemical indexes related to glucose metabolism. Histopathological examination manifested that liposomes alleviated liver inflammation in mice and showed a protective effect on liver. Furthermore, the anti-diabetic effect of HDCA:ME-(1:1)-Lips group was significantly better than that of the metformin group, which confirmed that HDCA played a synergistic role in the treatment of type 2 diabetic mice. HDCA:ME-(1:1)-Lips was the most effective among the three types of liposomes. These results indicated that HDCA was a potential liposome excipient for the treatment of type 2 diabetes, which provided a new pathway for clinical application of HDCA. However, there are still some limitations in the study. We only proved that HDCA-modified metformin liposomes could synergistically treat type 2 diabetes, but did not investigate other characteristics of liposomes, such as low toxicity, sustained-release, and high bioavailability. Therefore, our studies will focus on toxicity and tissue distribution experiments of liposomes to demonstrate their detoxification and sustained-release effects in the future.

## Figures and Tables

**Figure 1 molecules-28-02471-f001:**
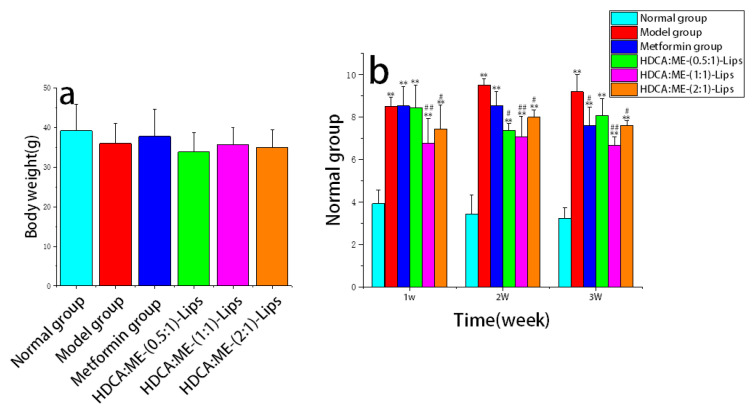
Effects of HDCA-ME-Lips on body weight (**a**) and food intake (**b**) in type 2 diabetic mice. All the values are represented as mean ± SD (*n* = 6), where * represents significant differences compared with the normal group; # represents significant differences compared with the model group (** *p* < 0.01, # *p* < 0.05, ## *p* < 0.01).

**Figure 2 molecules-28-02471-f002:**
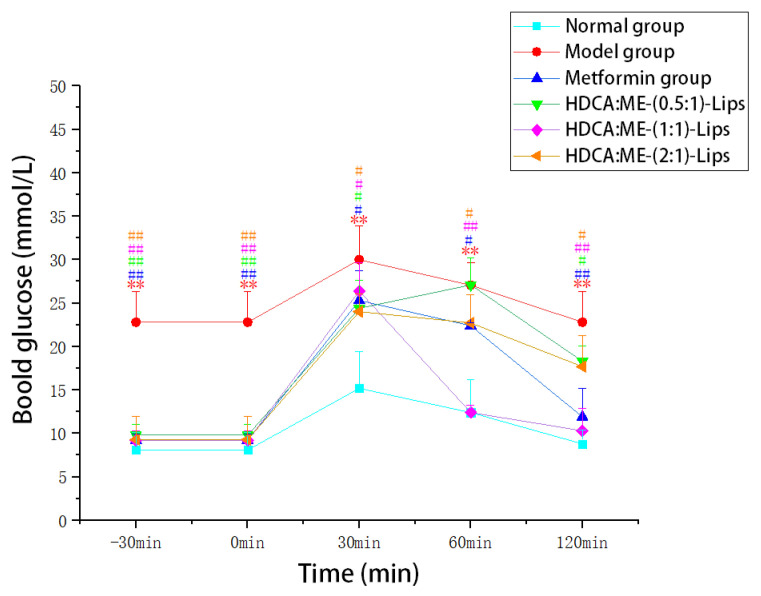
Effects of HDCA-ME-Lips on the blood glucose levels in oral glucose tolerance test (OGTT). All values are represented as mean ± SD (*n* = 6), where * represents significant difference compared with normal control and # represents significant differences compared with negative control (** *p* < 0.01, # *p* < 0.05, ## *p* < 0.01).

**Figure 3 molecules-28-02471-f003:**
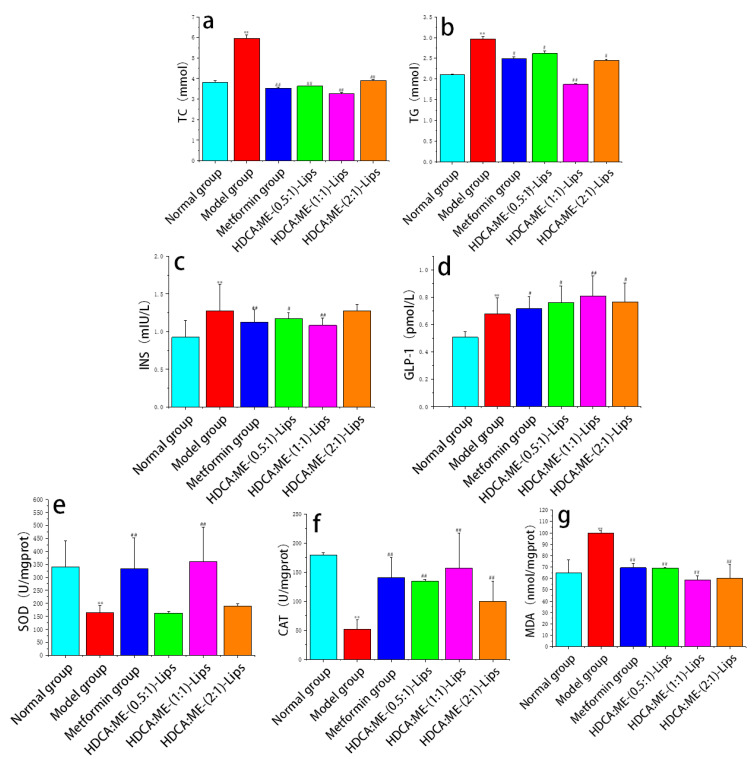
Effect of HDCA-ME-Lips on TC (**a**), TG (**b**), INS (**c**), GLP-1 (**d**), SOD (**e**), CAT (**f**), and MDA (**g**). All the values are represented as mean ± SD (*n* = 6), where * represents significant differences compared with normal control and # represents significant differences compared with negative control (** *p* < 0.01, # *p* < 0.05, ## *p* < 0.01).

**Figure 4 molecules-28-02471-f004:**
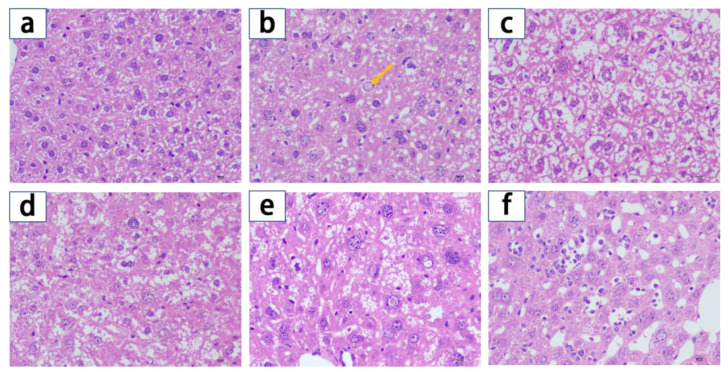
Liver histology images (HE staining 400×). (**a**) Normal group; (**b**) Model group; (**c**) Metformin group; (**d**) HDCA:ME-(0.5:1)-Lips group; (**e**) HDCA:ME-(1:1)-Lips group; (**f**) HDCA:ME-(2:1)-Lips group. Yellow arrow: pathological changes.

**Table 1 molecules-28-02471-t001:** Characterization of different liposomes (*n* = 3).

Liposomes	Particle Size	Polydispersity Index (PDI)	Encapsulation Efficiency (EE%)	Drug Loading (DL%)
HDCA:ME-(0.5:1)-Lips	196.36	0.259	67.15	5.52
HDCA:ME-(1:1)-Lips	232.30	0.256	61.28	5.35
HDCA:ME-(2:1)-Lips	332.40	0.258	59.53	5.16

**Table 2 molecules-28-02471-t002:** Effects of HDCA-ME-Lips on the blood glucose levels in diabetic mice.

Group	Concentrations of Fasting Blood Glucose(mmol/L)			
	D_0_	D_7_	D_14_	D_21_
Normal group	6.67	5.16	7.55	6.65
Model group	18.26 **	19.08 **	21.3 *	20.48 *
Metformin group	18.44 **	15.24 ^##^	15.73 ^#^	12.42 ^##^
HDCA:ME-(0.5:1)-Lips	18.42 **	13.08 ^#^	13.22 ^##^	12.34 ^##^
HDCA:ME-(1:1)-Lips	18.68 *	14.94 ^#^	14.64 ^#^	11.62 ^#^
HDCA:ME-(2:1)-Lips	18.8 *	15.88 ^##^	14.6 ^#^	15.8 ^##^

All the values are represented as mean (*n* = 6), where * represents significant differences compared with the normal group; # represents significant differences compared with the model group (* *p* < 0.05, ** *p* < 0.01, ^#^
*p* < 0.05, ^##^
*p* < 0.01).

**Table 3 molecules-28-02471-t003:** Effect of HDCA-ME-Lips on biochemical indices (*n* = 6).

Biochemical Indices	HDCA:ME-(0.5:1)-Lips	HDCA:ME-(1:1)-Lips	HDCA:ME-(2:1)-Lips
TC	3.63 **	3.28 **	3.91 **
TG	2.62 *	1.87 **	2.44 *
INS	1.169 *	1.082 **	1.277
GLP-1	0.761 *	0.807 **	0.764 *
SOD	162.18	360.72 **	188.65
CAT	134.66 **	156.52 **	99.56 **
MDA	69.02 **	58.70 **	6039 **

All the values are represented as mean (*n* = 6), where * represents significant differences compared with the model group (* *p* < 0.05, ** *p* < 0.01).

## Data Availability

The data presented in this study are available upon request from the corresponding author.

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
