# Peer review of "A Novel Drug Delivery System: Hyodeoxycholic Acid-Modified Metformin Liposomes for Type 2 Diabetes Treatment"

_molecules, 2023, doi:10.3390/molecules28062471_

Round 1

Reviewer 1 Report

In this manuscript, Hu and Gou et al. designed a new form of metformin with hyodeoxycholic acid and conducted some preliminary studies in mice to prove its glucose-lowering effect. The drug design is fascinating and of substantial significance. However, I would like to provide some comments that hopefully could be helpful in improving the study:

1. Is there any data about the changes in body weight and food intake during the treatment? Since metformin has been shown to reduce body weight, bile acid also potentially affects food intake and body weight.

2. There is no baseline oral glucose tolerance test data for all groups. It would be better to see the changes in blood glucose in each group.

3. Metformin and bile acids could augment the secretion of gastrointestinal hormones (e.g. GLP-1) which are key drivers in postprandial metabolic homeostasis (PMID: 31468642 and 33800566). I would suggest the authors measure the GLP-1 levels in response to treatments.

4. Since the drug is a combination of HDCA and metformin, it would be interesting to know whether the bile acid system is affected. It looks like the HDCA-Metformin has liver-protecting effect. Did the authors have any data to quantify the liver-protecting effect (e.g. PCR)

Minor issues:

1. The number of figures: figure 1 is not mentioned and showed in the manuscript.

2. In figure 2, there is no error bar. Please add the error bar in the revised version.

3. The words in figure 3 are too small. Hard to read.

Reviewer 2 Report

The authors studied the anti-diabetic effect of HDCA Metformin-liposome in STZ mice to illustrate the potential for Metformin in a liposomal formulation in lowering blood glucose and other parameters in the mice model. This study suffers with several limitations, including the experimental design and clarity of the data. The following observations were made:

  • The authors demonstrate the HDCA Metformin (1:1) formulation lowered blood glucose levels comparably better than the other two formulations in Fig2. Did the authors statistically compare the anti-diabetic effects between the three formulations in the oral glucose tolerance test and other parameters? How would the authors account for the observed differences in anti-diabetic effect amongst the different parameters? This is particularly relevant since the encapsulation efficiency was higher in the HDCA Metformin (0.5:1) than in the other two formulations in Table 1.
  • The authors report a high standard deviation in the means of levels of different biochemical parameters, such as SOD, CAT etc. in certain experimental groups, as presented in Fig 3. The analysis of the statistical significance of experimental data becomes tedious in such cases.
  • The representation of H&E staining of liver tissue sections in Fig 4 is accompanied by a description of the pathological changes observed in the experimental groups. However, it would have been helpful if the authors had indicated those changes in the relevant figure, the pathological changes they describe. Do the authors observe any noticeable changes in the liver architecture between the three liposomal formulations?

Reviewer 3 Report

In this paper “The study of the treatment of type 2 diabetic mice via hyodeoxycholic acid-modified metformin liposomes the authors analyzed a liposome-based delivery system. Liposomal delivery offers a targeted and almost complete absorption of the active substance. This method is more and more used in recent years due to their advantages, mostly the ability to assist the medication from potential damaging factors. The active ingredient is encapsulated in a phospholipid layer to protect it against the acidity in the stomach and to be transported, delivered, and absorbed, thus the bioavailability rates are much improved. In other delivery methods, the active ingredient loses most of its potency while passing through the GI tract and it is not absorbed in the small intestine. In liposomal delivery systemsthe bioavailability rates of the active substances are much higher, thus, the effect on the body is higher.

In this experiment, several liposome-based delivery system were used for metformin, a known anti-diabetic medication. Hyodeoxycholic acid, a cholesterol-like structure, was used in different proportions with metformin, adapting thin-film dispersion method, to find the right amount of hyodeoxycholic acid. In mice with type 2 diabetes, the best formula reduced fasting blood glucose levels, improved glucose tolerance, regulated oxidative stress markers and protected liver tissues.

The topic is important and the manuscript provides a good analysis of the subject. However, it needs major improvements in presentation. Also, the level of English throughout your manuscript does not meet the required standard, some ideas are hard to comprehend, there are miss-spellings in figures... You may wish to ask a native speaker to check your manuscript for grammar, style and syntax.

I suggest to choose a better title 

The information about the animals is not complete: it is not clear the number of mice used; does the experimental protocol approval have a number?

Conclusion - this section is missing. Authors should summarize the results, state the limitations of their study, as well as the scope for future research.

Round 2

Reviewer 2 Report

The Authors have revised the manuscript thoroughly and carefully based on all the comments raised by me. 

Reviewer 3 Report

The authors addressed all the suggestions. The manuscript is much improved.

I have no further comments.